# Molecular epidemiology and genetic diversity of *Anaplasma* and *Theileria* spp. in Pakistani sheep

Wajiha Shahzadi[1☯], Mughees Akbar[1☯], Arooj Ijaz[1], Arshad Hussain[1], Umair Shabbir[1], Ioannis A. Giantsis[2], Shakir Ullah[3], Maged A. AlGaradi[4], Adil Khan[5], Furhan Iqbal[1]*, Ayman A. Swelum[4]*

1 Institute of Zoology. Bahauddin Zakariya University, Multan, Pakistan, 2 Department of Animal Science, Faculty of Agriculture, Forestry and Natural Environment. Aristotle University of Thessaloniki, Thessaloniki, Greece, 3 Department of Zoology, Abdul Wali Khan University Mardan Pakistan, Mardan, Pakistan, 4 Department of Animal Production, College of Food and Agriculture Sciences, King Saud University, Riyadh, Saudi Arabia, 5 Department of Zoology, Bacha Khan University, Charsadda, Khyber Pakhtunkhwa, Pakistan

☯ These authors contributed equally to this work.
* furhan.iqbal@bzu.edu.pk (FI); aswelum@ksu.edu.sa (AS)

## Abstract

Pakistan has a huge sheep population (37.2 million in 2024) that is largely unexplored for the presence of vector transmitted parasites. Present study was aimed to document the prevalence of *Anaplasma* sp*.*, *Anaplasma ovis*, *Theileria ovis* and *Theileria lestoquardi* in sheep blood samples (N = 329) that were collected from six districts (Muzaffargarh, Rajanpur, Dera Ghazi Khan, Layyah, Taunsa and Khanewal) during August till December 2024 and to report the genetic diversity of screened pathogens. Molecular analyses revealed that the prevalence of *Anaplasma* sp*.*, *Anaplasma ovis* and *Theileria ovis* in screened sheep was 11%, 20% and 21% respectively. None of the screened sheep was *Theileria lestoquardi* infected. Co-infection of the screened pathogens was also observed. Presence of the detected pathogens was confirmed by DNA sequencing and subsequent BLAST analysis. Phylogenetic analysis revealed that these pathogens displayed genetic similarities with the sequences that were deposited from various countries across the globe. Prevalence of all screened pathogens varied significantly between the sampling districts. Similarly, the *Anaplasma* sp., *Anaplasma ovis* and *Theileria ovis* prevalence varied significantly among the sheep breeds. *Anaplasma ovis* infection was more common in large herds and in un-infested sheep. *Theileria ovis* infection was more frequent in small herds. In conclusion, we are reporting the presence of *Anaplasma* sp., *Anaplasma ovis* and *Theileria ovis* in Pakistani sheep that were enrolled from all six districts. Large-scale studies are recommended in various geo-climatic regions of Pakistan to confirm the genetic diversity, epidemiology and host-pathogen interactions that will contribute towards effective control of these infections among the local sheep population.

**Data availability statement:** https://www.ncbi.nlm.nih.gov/nuccore/PV274865, https://www.ncbi.nlm.nih.gov/nuccore/PV274866, https://www.ncbi.nlm.nih.gov/nuccore/PV274867, https://www.ncbi.nlm.nih.gov/nuccore/PQ855658, https://www.ncbi.nlm.nih.gov/nuccore/PQ855659, https://www.ncbi.nlm.nih.gov/nuccore/PQ836114, https://www.ncbi.nlm.nih.gov/nuccore/PQ836115, https://www.ncbi.nlm.nih.gov/nuccore/PQ836116.

**Funding:** The authors extend their appreciation to the Ongoing Research Funding Program (ORF-2025-971), King Saud University, Riyadh, Saudi Arabia, for funding this research. The funders had no role in study design, data collection and analysis, decision to publish, or preparation of the manuscript.

**Competing interests:** The authors have declared that no competing interests exist.

## 1. Introduction

In an agricultural country like Pakistan, economy heavily relies on the livestock sector. Trading animals and their products is the major source of income especially in semi-urban and rural areas of this country [1]. Small ruminant farming is major part of livestock sector in this country meeting the meat, milk and hides requirements of the local population. Pakistan's sheep population is estimated to be 26.5 million with major breeds including Balkhi, Salt Range, Kaghani, Damani, Baluchi, Hastnagri, Lohi, Thalli and Kachhi [2]. Pakistan ranks third globally after China and India in terms of sheep and goat population. However, the economic returns from this substantial livestock resource remain far below expectations [3]. One of the key limiting factors is the widespread prevalence of ticks and tick-borne diseases (TBDs), which pose a serious threat to livestock health and productivity [4]. This challenge is particularly severe in tropical and subtropical regions like Pakistan, where environmental conditions favor tick survival and reproduction [5]. Tick infestations and TBDs are estimated to cause annual losses of approximately USD 200 million in Pakistan's livestock sector. Compounding the problem is the emergence of acaricide-resistant tick populations, which has further complicated effective management and control efforts [4]. Theileriosis is a common tick-borne infection that is caused by various *Theileria* species: protozoan parasites [6]. While anaplasmosis is caused by bacterial species that belongs to *Anaplasma* genera and they are known to affect small ruminants globally resulting in massive economic losses [7].

Anaplasmosis affects wild and domestic animals and it has been reported from Mediterranean Basin, central Europe as well as from the tropical and sub-tropical regions of the world [8]. A variety of Anaplasma species infect sheep including Anaplasma marginale, *Anaplasma* ovis, Anaplasma phagocytophilum and *Anaplasma* capra [4]. A variety of ticks species belonging to Haemaphysalis , Ixodes and Rhipicephalus genera are transmit these bacteria to sheep and goats [9]. In small ruminants, this infection results in reduced milk yield, weight loss, anemia and abortion [8]. Usually small ruminants are treated with intramuscular injection of 20 mg/Kg of oxytetracycline to treat anaplasmosis [10].

An apicomplexan parasites *Theileria lestoquardi* that is transmitted to small ruminants by the *Hyalomma* sp. ticks causes malignant ovine theileriosis that causes fever, cough, lethargy, lymphadenopathy and weight loss and can lead to mortality in case of severe infection [11]. Infection with *Theileria ovis* results in benign theileriosis in small ruminants that results in weight loss, fever and decreased production but it is generally less severe than the infection caused by *Theileria lestoquardi* [1]. Ixodid ticks, especially those belonging to genera *Rhipicephalus*, are known to transmit *Theileria ovis* to various animals [2]. Anti *Theileria* drugs, Parvaquone and Buparvaquone, are used to treatment theileriosis in small ruminants [12].

Sheep are playing an important role in common man's life in Pakistan but screening blood borne infectious agents among them is uncommon. To address this gap, sheep were enrolled from six districts in Punjab (Pakistan) and their blood samples were screened for the presance of *Anaplasma* sp., *Anaplasma* ovis, *Theileria ovis* and *Theileria lestoquardi* through PCR and DNA sequencing approach. The

pathogens selected for the screening are frequently reported in local small ruminant but sheep from five of the six districts included in this investigation has never been screened for these bacteria and parasites previously. Additionally, epidemiology and genetic diversity of these pathogens was also investigated.

## 2. Materials and methods

### 2.1. Study area and blood sampling

Molecular prevalence of *Anaplasma* sp., *Anaplasma ovis, Theileria ovis* and *Theileria lestoquardi* among sheep was investigated through an epidemiological survey that was conducted sp. in six districts [Khanewal (30.3000°N and 71.9333°E), Rajanpur (29.104650°N and 70.325665°E), Muzaffargarh (30.0736°N and 71.1805°E), Taunsa (30.7056°N and 70.6578°E), Dera Ghazi Khan (30.0499°N and 70.6455°E) and Layyah (30°57′55" N and 70°56′38" E)] of Punjab. We hypothesized that parasitic infection rates will be diverse among the sheep as they were enrolled from areas that were geo-climatically distinct (Fig. 1). Ethical Research Committee of the Bahauddin Zakariya University Multan (Pakistan) approved all the experimental procedures and protocols applied in this study via letter number BZU./ Ethics/24–37. A total of 329 sheep blood samples belonging to 7 different breeds: Desi, Mundri, Kajli, Bluchi, Lohi, Damani and Balkhi breeds were randomly collected following the informed consent of sheep owners during August till December 2024. We used Solvin's formula to estimate sample size during during present study. Solvin's formula was computed as; n = N/ (1 + N * e2). Whereas: n = no. of samples, N = total population, e = margin of error [11]. All the rnrolled animals were apparently healthy and reared as heard with variable number of heads. Herds having up to one hundred small ruminants were considered small while those having more than one hundred heads were considered as large herds. The enrolled sheep includes

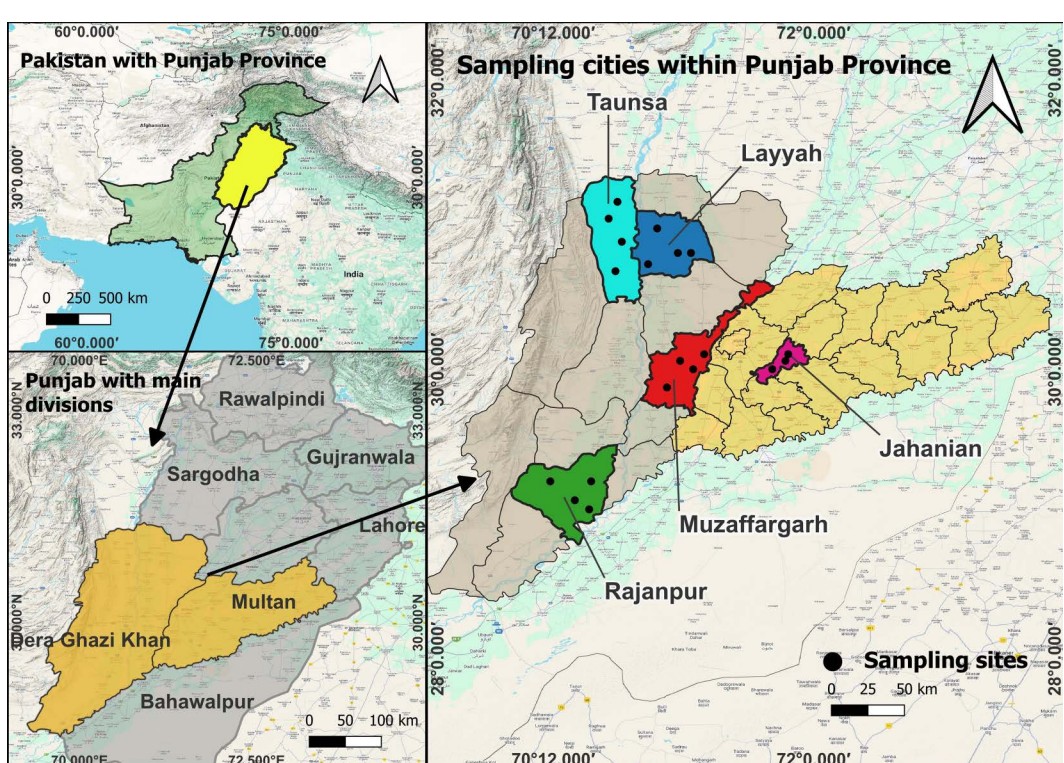

**Fig 1. Map of Pakistan showing the Punjab province in yellow.** The magnified maps of Punjab are presenting the districts in various colors from where the sheep blood samples were collected during present investigation.

49 males (15%) and 280 females (85%). Sheep were enrolled from six districts: Khanewal (N = 100), Rajanpur (N = 45), Muzaffargarh (N = 32), Taunsa (N = 50), Dera Ghazi Khan (N = 51) and Layyah (N = 51). Two to three milliliter blood was sampled in EDTA coated labelled collection tubes following aseptic Jugular vein prick to each animal. A questionnaire was filled for each animal with the help of livestock owner, during blood sample collection, in order to collect basic epidemiological data (sampling site, date of sample collection, sex, breed, ectoparasites, herd size and dogs with herds) associated with the prevalence of pathogens that were screened among the sheep.

## 2.2. DNA extraction and detection of pathogens by PCR

An inorganic shock method, as previously reported by Aziz et al. [4], was followed for genomic DNA extraction from sheep blood. Gel electrophoresis was used to confirm the DNA and then it was screened for the detection of *Anaplasma* sp. (target was 16S rRNA gene), *Anaplasma ovis* (*msp4* gene was targeted), *Theileria ovis* (18S rRNA gene was amplifies) and *Theileria lestoquardi* (18S rRNA gene was targeted) by using previously reported primers and protocols [13–16] (Supplementary Table 1). Distilled water, *Anaplasma ovis Theileria ovis* positive DNA (previously confirmed through Sanger sequencing) were amplified as negative and positive controls during each PCR, respectively.

## 2.3. DNA sequencing and phylogenetic analysis

Amplified PCR products were sent to a commercial lab (First Base Malaysia) for DNA sequencing. The resultant sequences were viewed and edited in FinchTV (version 1.4.0). Low-quality nucleotides were removed from both ends of the sequences. BLAST analysis was performed for the sequence confirmation and was and subsequently deposited in GenBank. Pathogenic sequences, similar to the genes targeted in this stud, were downloaded from BLAST output and they were used in phylogenetic analysis. Sequences were aligned through ClustalW integrated model selection tool of MEGA was used for best fit model selection. Finally, Maximum Likelihood method with 1000 bootstraps was used to infer the phylogenetic tree. 16S rRNA gene of an *Ehrlichia canis, msp4* gene of *Rickettsia hoogstraalii* and 18S rRNA gene of *Babesia microti* were used as outgroup during the phylogenetic analysis of *Anaplasma* sp., *Anaplasma ovis* and *Theileria ovis* respectively.

## 2.4. Statistical Analysis

Data was analyzed by using Minitab (Minitab, USA). Values equal to or less than 0.05 were considered statistically significant. Studies risk factors were correlated with pathogenic prevalence by using Fischer exact test. One way ANOVA was applied to compare parasite prevalence between sample collection sites as well as between sheep breeds. Chi square test was applied to compare the prevalence of all pathogens between the screened sheep.

## 3. Results

### 3.1. Molecular epidemiology of *Anaplasma* sp. in sheep blood samples

PCR amplified 345 base pairs fragment from the 16S rRNA gene of *Anaplasma* sp. in 35 out of 329 (11%) sheep blood samples that were collected during present study (Table 1). Sheep enrolled from all six sampling districts were infected with *Anaplasma* sp. Results of one-way ANOVA test indicated that the prevalence of *Anaplasma* sp. varied significantly between the sampling district (P = 0.007). The highest bacterial prevalence was detected in sheep from Taunsa district (20%) followed by Rajanpur (18%), Dera Ghazi Khan (12%), Layyah (10%), Khanewal (9%) and Muzaffargarh (3%) (Table 2).

During the present investigation, seven sheep breeds were enrolled and *Anaplasma* sp. infection was detected in all of them. One-way ANOVA results revealed that the pathogen prevalence varied significantly when compared between the sheep breeds (P = 0.002). The highest prevalence was found in Balkhi sheep (42%), followed by Kajli (20%), Damani (16%),

**Table 1. Comparison of *Anaplasma* spp.*, Anaplasma ovis* and *Theileria ovis* prevalence among the sheep enrolled from six districts in Punjab along with the co infection data generated during present study. % prevalence of each pathogen is given in parenthesis. P –value indicates the results of Chi-square test calculated to compare the prevalence of two bacteria among the screened animals.**

| *Anaplasma* spp. + ve | *Anaplasma ovis* +ve | *Theileria ovis* +ve | X² | P value | Co infection | | | |
|---|---|---|---|---|---|---|---|---|
| | | | | | *Anaplasma* spp. and *Anaplasma ovis* | *Anaplasma* spp. and *Theileria ovis* | *Anaplasma ovis* and *Theileria ovis* | All three pathogens |
| 35/329 (11%) | 63/329 (20%) | 70/329 (21%) | 14.763 | 0.001 *** | 10/329 (3%) | 10/329 (3%) | 23/329 (7%) | 4/329 (1%) |

P < 0.001(***) = Highly significant.

**Table 2. Overall comparison of *Anaplasma* spp.*, Anaplasma ovis* and *Theileria ovis* prevalence among the sheep enrolled from six districts in Punjab during the present study. % prevalence is given in parentheses. P-value represents the output of one -way ANOVA test.**

| Sampling district | *Anaplasma* spp. + ve | P value | *Anaplasma ovis* +ve | P value | *Theileria ovis* +ve | P value |
|---|---|---|---|---|---|---|
| Muzaffargarh | 1/32 (3%) | | 4/32 (13%) | | 33/100 (33%) | |
| Rajanpur | 8/45 (18%) | | 3/45 (7%) | | 12/45 (27%) | |
| Dera Ghazi Khan | 6/51 (12%) | | 9/51 (18%) | | 8/32 (25%) | |
| Layyah | 1/51 (10%) | 0.007 ** | 30/100 (30%) | P<0.001*** | 8/50 (16%) | 0.001 *** |
| Taunsa | 10/50 (20%) | | 5/51 (10%) | | 5/51 (10%) | |
| Khanewal | 9/100 (9%) | | 12/50 (24%) | | 4/51 (8%) | |
| **Total** | **35/329 (11%)** | | **63/329 (19%)** | | **70/329 (21%)** | |

P < 0.01 = Significant (*); P < 0.01 = Highly significant (***).

**Table 3. Prevalence of *Anaplasma* spp.*, Anaplasma ovis* and *Theileria ovis* among the screened sheep breeds enrolled from six sampling districts in Punjab during the present study. % prevalence of pathogen is given in parentheses. P-value represents the results of the one way ANOVA test.**

| Breed | *Anaplasma* spp. + ve | P value | *Anaplasma ovis* +ve | P value | *Theileria ovis* +ve | P value |
|---|---|---|---|---|---|---|
| Mundri | 09/59 (15%) | | 8/59 (14%) | | 33/100 (33%) | |
| Desi | 03/98 (3%) | | 12/98 (12%) | | 8/25 (32%) | |
| Kajli | 05/25 (20%) | | 6/25 (24%) | | 2/7 (29%) | |
| Damani | 03/18 (16%) | | 2/18 (11%) | | 4/22 (18%) | |
| Balochi | 03/22 (13%) | 0.002** | 1/22 (05%) | 0.001*** | 10/59 (17%) | 0.003** |
| Balkhi | 03/07 (42%) | | 4/07 (57%) | | 13/98 (13%) | |
| Lohi | 09/100 (9%) | | 30/100 (30%) | | 0/18 (0%) | |
| **Total** | **35/329 (11%)** | | **63/329 (19%)** | | **70/329 (21%)** | |

P < 0.01 = Significant (**); P < 0.001 = Highly significant (***).

Mundri (15%), Balochi (13%), Lohi (9%) and Desi (3%) breed (Table 3). Overall risk factor analysis revealed that the *Anaplasma* sp. prevalence was not associated with a particular sheep sex (P = 0.1) or with the presence or absence of ectoparasite on sheep (P = 0.6), presence of dogs with the herd (P = 1) or with the size of the screened herds (P = 0.7) (Table 4).

### 3.2. Phylogenetic analysis of 16S rRNA gene of *Anaplasma* sp

Three positive partial 16S rRNA gene sequences of *Anaplasma* sp. were confirmed by DNA sequencing. BLAST analysis revealed that the amplified sequences were 99–100% similar with the 16S rRNA sequences of various *Anaplasma*

Table 4. Overall association of the studied risk factors with the prevalence of *Anaplasma* spp*., Anaplasma ovis* and *Theileria ovis* among the sheep enrolled from six sampling districts in Punjab during the present study. Prevalence of pathogen (%) is given in parenthesis. P-value represents the results of Fischer's exact test calculated for each studied parameter.

| Parameters | | *Anaplasma* spp.+ve | P value | *Anaplasma ovis* +ve | P value | *Theileria ovis* +ve | P value |
|---|---|---|---|---|---|---|---|
| Sex | Female | 27/282 (09%) | 0.1 | 51/282 (18%) | 0.2 | 60/282 (21%) | 1 |
| | Male | 08/47 (17%) | | 12/47 (26%) | | 10/47 (21%) | |
| Ectoparasite | Present | 13/112 (11%) | 0.6 | 12/112 (11%) | 0.001*** | 19/112 (17%) | 0.2 |
| | Absent | 22/217 (10%) | | 51/217 (24%) | | 51/217 (24%) | |
| Dogs with herd | Positive | 26/184 (14%) | 01 | 31/184 (17%) | 0.2 | 34/184 (18%) | 0.2 |
| | Negative | 09/145 (06%) | | 32/145 (22%) | | 36/145 (25%) | |
| Herd size | Large | 29/70 (42%) | 0.7 | 19/70 (27%) | 0.05* | 42/259 (16%) | P<0.001*** |
| | Small | 06/259 (15%) | | 44/259 (17%) | | 28/70 (40%) | |

P>0.05=Non significant, P<0.001=Highly significant (***).

species (including *Anaplasma* sp., *A. marginale, A. capra, A. phagocytophilum, A. Platys* and *A. bovis*) that were previously deposited in GenBank. 16S rRNA gene sequences generated in this study was deposited in GenBank with accession numbers PV 274865, PV274866 and PV274867. Phylogenetic analysis revealed that the Pakistani isolates generated in this study were genetically similar as they all clustered together and they showed similarities with 16S rRNA sequences of *Anaplasma* sp. detected in mammals and ticks that were previously deposited from Iran (ON333753, ON333754 and MH879781), China (KX987331) and Morocco (OK606076), *Anaplasma marginale* in large ruminants reported from Taiwan (OL660542 and OL660543), *Anaplasma capra* in ticks from South Korea (MT052416 and MT052417), *Anaplasma phagocytophilum* in rodents from South Africa (MK814411 and MK814412), *Anaplasma platys* in ticks from China (OK639187 and OK639188) and *Anaplasma bovis* in ruminants from China (JN558819) and Japan (AB588969) (Fig. 2).

### 3.3. Molecular epidemiology of *Anaplasma ovis* in sheep blood samples

PCR amplified a 347 base pairs fragment specific for *msp4* gene of *Anaplasma ovis* in in 63 out of 329 (20%) sheep blood samples that were collected during present study (Table 1). Analysis of the data revealed that *Anaplasma ovis* was detected in sheep from all six districts that were included in this study. One-way ANOVA results indicated that the bacterial prevalence in sheep varied significantly among the sampling districts (P=0.01). The highest infection rate was observed in sheep enrolled from Khanewal (30%) followed by Taunsa (22%), Dera Ghazi Khan (18%), Muzaffargarh (13%), Layyah (10%) and Rajanpur (7%) (Table 2).

All seven sheep breeds that were enrolled during this investigation were *Anaplasma ovis* infected. One- way ANOVA test results revealed that the bacterial prevalence varied significantly (P=0.001) among the screened sheep breeds. The highest prevalence of *Anaplasma ovis* was observed in Balkhi (57%) followed by Lohi (30%), Kajli (24%), Mundri (14%), Desi (12%), Damani (11%) and Balochi (5%) breed (Table 3). Overall risk factor analysis revealed that the *Anaplasma ovis* infection was more common in large herds as compared to smaller ones (P=0.05). Also sheep without ectoparasite infestation had significantly higher *Anaplasma ovis* infection then tick infested sheep (P=0.001). Fisher exact test results revealed that bacterial prevalence was not associated with a specific sheep sex (P=02 or with the presence of dogs with sheep herds (P=02) (Table 4).

### 3.4. Phylogenetic analysis of *msp4* gene of *Anaplasma ovis*

Two randomly selected *Anaplasma ovis* positive PCR products were confirmed by DNA sequencing and deposited in GenBank with accession numbers PQ855658 and PQ855659. These Pakistani isolates were genetically similar as they were

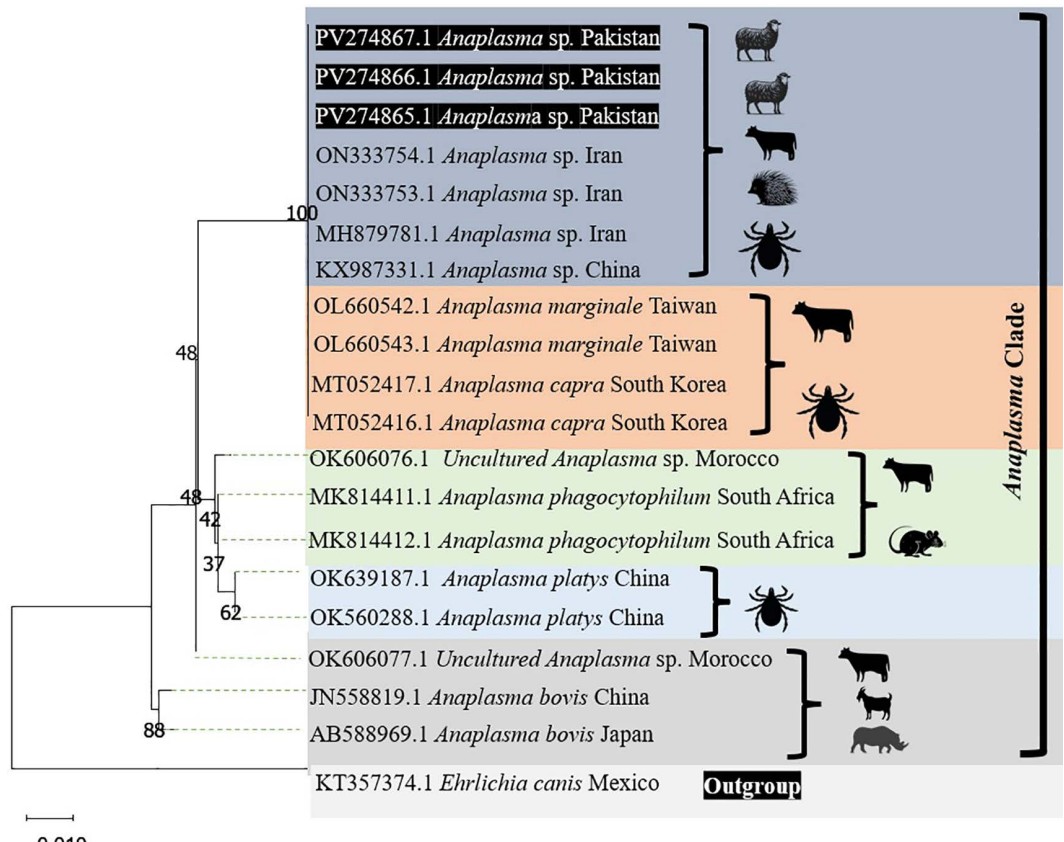

**Fig 2. Phylogenetic analysis of *Anaplasma* sp. based on the partial 16S rRNA gene sequence.** The three haplotypes (PV274865, PV274866 and PV274867) generated in this study are highlighted in red.

clustered together and they displayed similarities with the *msp4* gene sequences of *Anaplasma ovis* that were previously deposited from Pakistan (PQ553182, PQ553183 and OP620759), China (KY511046), Thailand (MK140725, MK140732, and KU764497), Hungary (EF190512 and EF190513), Turkey (MT344081, MT344082, OQ167969 and OQ167970), Uganda (MT247053) Serbia (GQ925818 and GQ925819), South Africa (KY305598) and Iran (MH549704). Pakistani *Anaplasma ovis msp4* gene sequences were distinct from *Anaplasma marginale* sequences reported from South Africa and Thailand. As well as they were distant from the partial msp4 sequences of *Rickettsia hoogstraalii* (MH549704) isolated in Iran and used as an outgroup during this phylogenetic analysis (Fig. 3).

### 3.5. Molecular epidemiology of *Theileria ovis* in sheep blood samples

Polymerase chain reaction amplified a 520 base pairs fragment from the 18S rRNA gene of *Theileria ovis* in 70 out of 329 (21%) sheep blood samples that were collected during present study (Table 1). Analysis of the data revealed that sheep enrolled from all six sampling districts were infected with *Theileria ovis*. One-way ANOVA test results indicated that the parasite prevalence in sheep varied significantly among the sampling districts (P = 0.001). The highest infection rate was observed in sheep enrolled from Khanewal (33%) followed by Rajanpur (27%), Muzaffargarh (25%), Taunsa (16%), Dera Ghazi Khan (10%) and Layyah (8%) (Table 2).

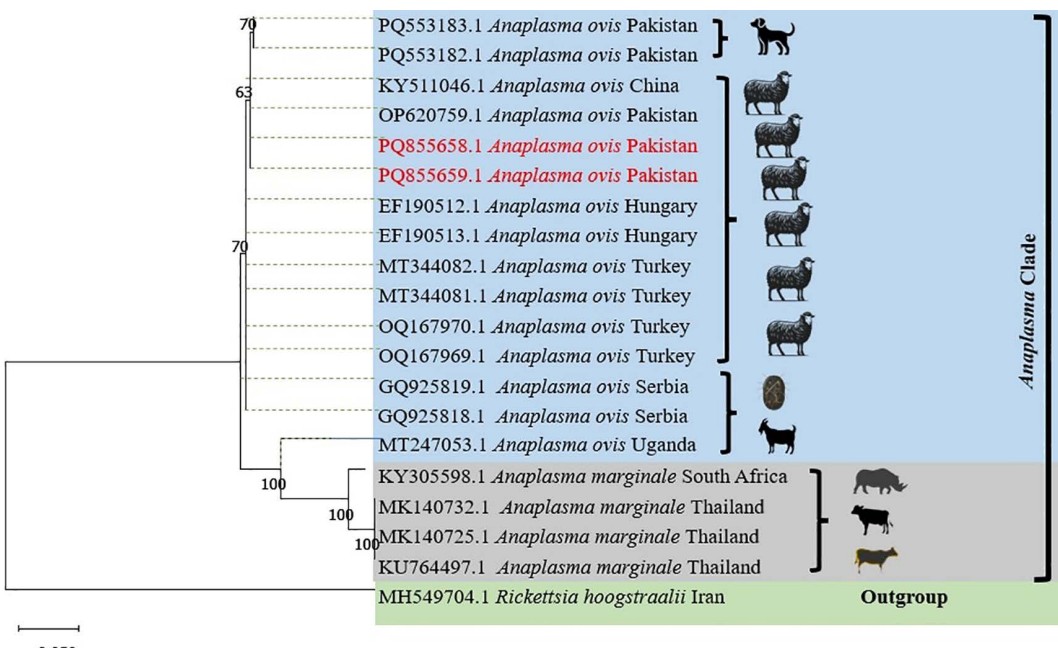

**Fig 3. Phylogenetic tree of *Anaplasma ovis* based on the partial *msp4* gene sequences available in GenBank.** The two haplotypes (PQ855658 and PQ855659) generated in this study are highlighted in red.

Six out of seven sheep breeds (except Damani) enrolled in this study were infected with *Theileria ovis*. One-way ANOVA test results revealed that the pathogen prevalence in sheep varied significantly among the screened sheep breeds (P = 0.003). The highest prevalence of *Theileria ovis* was observed in Lohi breed (33%) followed by Kajli (32%), Balkhi (29%), Baluchi (18%), Mundri (17%) and Desi (13%) breed (Table 3). Overall risk factor analysis revealed that the *Theileria ovis* infection was more common in small herds as compared to large ones (P < 0.001) but presence or absence of ectoparasites on sheep (P = 0.2), presence of dogs with the herds (P = 0.2) as well as sheep sex (P = 1) did not influenced parasite prevalence during present investigation (Table 4).

### 3.6. Phylogenetic analysis of 18S rRNA gene of *Theileria ovis*

Three randomly selected *Theileria ovis* positive PCR products were confirmed by DNA sequencing and deposited in GenBank with accession numbers PQ836114, PQ836115 and PQ836116. Phylogenetic analysis confirmed that these Pakistani isolates were genetically similar as they were clustered together and they displayed similarities with the 18S rRNA gene sequences of *Theileria ovis* that were previously deposited from India (MT418209, MT418210, MZ220429 and M220430), Ethiopia (KF557876 and KF557877), Turkey (EF092453, KT851429, KT851430, PP463887 and PP463888), Egypt (OP389058, OP389059 and MN625903) and France (EU622911). Pakistani *Theileria ovis* 18S rRNA gene sequences were distinct from *Theleria annulata* (reported from Pakistan and Iraq), *Theileria lestoquardi* (reported from Egypt, *Theileria orientalis* from South Africa and China) as well as from *Theileria buffeli* sequences deposited from China. Furthermore, they were distant from the partial 18S rRNA gene sequences of *Babesia microti* (PQ538588)) isolated in Pakistan and used as an out-group during this phylogenetic analysis (Fig. 4).

### 3.7. Molecular epidemiology of *Theileria lestoquardi* in sheep blood samples

None of the sheep screened during this investigation was found infected with Theileria lestoquardi. Hence, data analysis for this parasite was not possible.

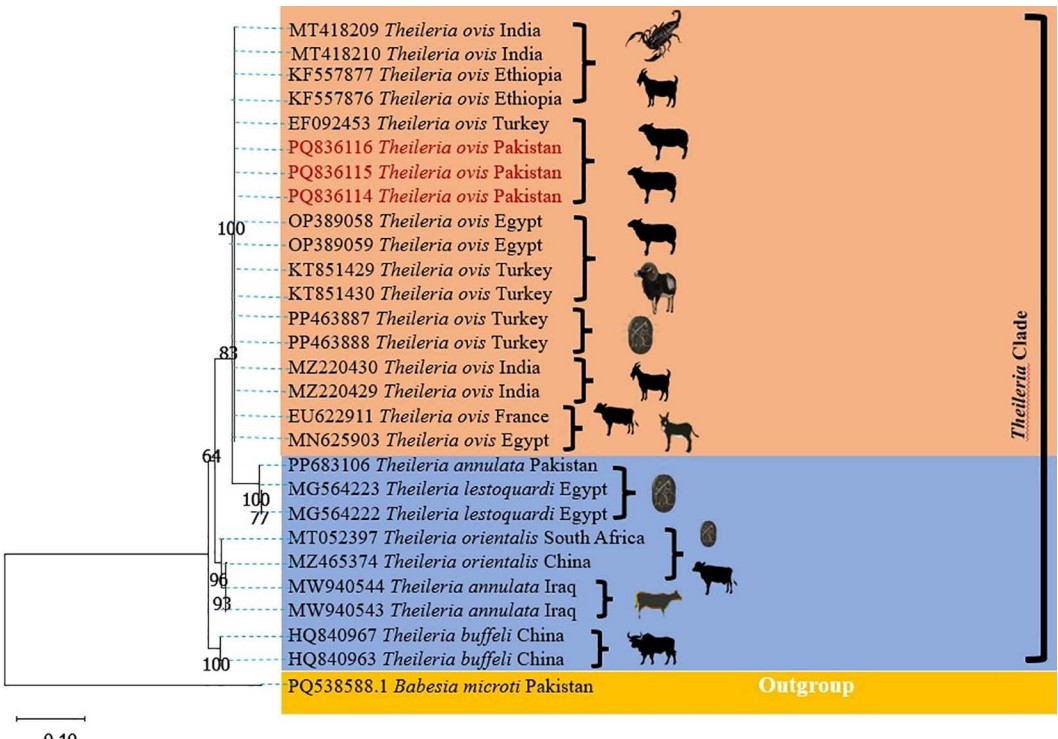

**Fig 4. Phylogenetic tree of *Theileria ovis* based on the partial 18S rRNA gene sequences available in GenBank.** The three haplotypes (PQ836114, PQ836115 and PQ836116) generated in this study are highlighted in red.

### 3.8. Co infection of pathogens among the sheep blood samples

Prevalence of the three screened pathogens varied significantly among enrolled sheep (P = 0.001). *Theileria ovis* (21%) had the highest prevalence among the screened sheep followed by *Anaplasma ovis* (20%) and *Anaplasma* spp. (11%). Co infection with two parasites was also observed. Overall, 23 sheep (7%) were co-infected with *Anaplasma ovis* and *Theileria ovis*, 10 sheep were co-infected either with *Anaplasma* spp. *and Theileria ovis* (3%) or with *Anaplasma* spp. and *Anaplasma ovis* (3%). Four sheep (1%) were simultaneously infected with all three pathogens that were screened during present investigation (Table 1).

## 4. Discussion

Sheep are known to easily adopt to various living conditions and their raising cost is low that makes them a popular live-stock among people in rural areas of Pakistan having less income [6]. Due to their grazing habit, sheep are more exposed to the environment and vectors. So, they have more chance to acquire infections, act as a infection reservoir and to be involved in their onward transmission [10]. Small ruminants annually move between pastures during summer and winter seasons pastures and hence they are big source in the transmission of ticks and tick-borne infections [17]. Research on these issues remained uninvestigated in many countries including Pakistan. Hence, this investigation was aimed to screen four common blood borne pathogens among the local sheep breeds.

During present study, we observed that 11% of the enrolled sheep were infected with *Anaplasma* spp. We used the generic primers during this investigation, so more than one bacterial species were expected during this screening. A potential reason for the success of this bacterium is probably the fact that they are transmitted to ruminants by a variety

of hard tick species [9]. During present investigation, we have targeted 16S rRNA gene of *Anaplasma* spp. for the phylogenetic analysis. This gene is a popular choice for phylogenetic analysis because it is ubiquitous in bacteria and archaea and it has both conserved and variable regions and its function is essential for all cells. These properties make them a suitable candidate not only for broad phylogenetic comparisons but also for more detailed intra specific analysis [18]. Three amplicons from the 16S rRNA gene of this bacterium were included in the phylogenetic analysis and they showed similarities with the 16S rRNA gene sequence of *Anaplasma* sp. reported in cattle [19] and long eared hedgehogs [20] in Iran, ticks in China [21] and cattle in Morocco (unpublished data). The sequences generated in this study also showed marked similarities with *Anaplasma marginale* that was detected in ticks infesting small mammals in Taiwan [22] as well as with *Anaplasma capra* that were detected in ticks infesting cattle in South Korea [23], *Anaplasma phagocytophilum* reported from rodents in South Africa [24], *Anaplasma platys* detected in ticks from China (unpublished data) and 16S rRNA sequence of *Anaplasma bovis* reported from goats in China [25] and wild boars and deer in Japan [26]. Future phylogenetic studies are required in small ruminants for the better understanding of the genetic diversity of *Anaplasma* sp. among the local sheep and goats.

During present study, we observed a relatively high prevalence of *Anaplasma ovis* (20%) among the sheep that were enrolled from six districts in Punjab. *Anaplasma ovis* infection among Pakistani sheep has been reported previously. Infection of this bacterium in sheep was 49.5% in Dera Ghazi Khan and Jhang districts [27], 26% in Fort Munro [28], 12.5% in Dera Ghazi Khan district [3] in Punjab. While Niaz et al. [29] had found that 21.75% of sheep and goats enrolled from Malakand, Swat, Bajaur and Shangla districts in Khyber Pakhtunkhwa were *Anaplasma ovis* infected. The prevalence of *Anaplasma ovis* in sheep has been reported from various parts of the world. Prevalence of this bacterium was 80% [30] and 62% in Tunisia [31], 69% in small ruminants of Mongolia [32], 54.5% in China [33], 34.2% in central and Western Kenya [34], 29.7% in Turkey [35], 20.8% in Iran [36], 14.8% in Bangladesh [37] and 1.5% in Thailand [38]. There is a clear variation in the bacterial prevalence among the studies discussed above. A number of factors can influence infection rate of *Anaplasma ovis* in small ruminants including different geo-climatic conditions of sampling sites, age, sex and immunity levels of screened animals as well as vector abundance in the studied area and the type of farm management [39].

There are few reports available in literature from Pakistan regarding the genetic diversity of *Anaplasma ovis* detected in small ruminants, especially in sheep. Hence, both the amplified PCR products during present investigation from the *msp4* gene were used for the phylogenetic analysis. The *msp4* gene is used for phylogenetic analysis of *Anaplasma* species as it is part of the MSP2 protein superfamily and its sequences reveal genetic variations and allow for the characterization of different strains and their evolutionary relationships [40]. The *msp4* sequences generated in this study were similar to the *msp4* sequences of *Anaplasma ovis* isolated from sheep [3] and dogs (unpublished data) in Pakistan, sheep in Hungary [41], sheep and goats from Turkey [42] and small ruminants in Uganda and Serbia (unpublished data) (Fig. 3). These studies indicated that similar *Anaplasma ovis* sequences are globally present indicating that they likely belongs to the same *Anaplasma ovis* strain or genotype and this data will aid in understanding the pathogen's epidemiology and evolution that will lead towards its control [43].

During present study, we found that *Anaplasma* spp. and *Anaplasma ovis* varied significantly between the sampling sites and sheep breeds. Infection was common among large herds and un-infested sheep (Tables 2–4). Similar to what we observed, Atif et al. [27] also observed a significant variation in bacterial infection among sheep that were enrolled from Dera Ghazi Khan and Jhang districts in Punjab, Pakistan. They also found that age of host animal and acaricide use was also associated with *Anaplasma ovis* infection. Contrary to our results, Naeem et al. [3] found that sex, age, sheep breed, herd size and composition and presence of dogs with herd were not association with *Anaplasma ovis* infection in sheep that were enrolled from Dera Ghazi Khan District in Punjab. Niaz et al. [29] found that host, age, grazing system and acaricide treatment were significant determinants for *Anaplasma ovis* infection in small ruminants from Khyber Pakhtunkhwa in Pakistan but the bacterial infection was not limited to a particular sampling district. These variations in the risk

factor data warrants larger scale epidemiological surveys to be conducted in Pakistan as well as in countries worldwide for the better understanding of *Anaplasma ovis*-sheep interactions.

During present study, we observed a high prevalence of *Theileria ovis* (21%) among the sheep that were enrolled from various regions in Punjab. Prevalence of *Theileria ovis* among Pakistani sheep has been reported by a number of research groups. Infection of this parasite in sheep was 37% in Okara district [44], 18.44% in Lahore district of Punjab [45], 14.3% in Malakand, Swat, Bajaur and Shangla districts in Khyber Pakhtunkhwa [29], 10.6% in Layyah district [1], 6.1% in Rajanpur District [2], 6% in Multan district [46] and 3% in Fort Munro [28]. A major reason for the different *Theileria ovis* infection rate among these studies is the different geo-climatic conditions of the sampling sites that impacts the vector abundance and diversity and hence the prevalence of this parasite. In addition to climate, the farm management techniques and awareness of livestock owners regarding the ticks and other vectors also differ in these areas affecting the *Theileria ovis* prevalence among the investigations [47].

18S rRNA genes are commonly targeted to investigate the evolutionary history of vertebrates as they are relatively conserved with slow evolutionary rates [9]. 18S rRNA gene sequence amplified in this study showed similarities with the 18S rRNA gene sequence of *Theileria ovis* reported from Scorpion and small ruminants in India (unpublished data), domestic ruminants in Ethiopia [48], small ruminants [49] and ticks [50] in Turkey, large ruminants and equine in Egypt (unpublished data) and domestic animals in France [51]. This phylogenetic analysis is indicating that Theileria ovis can infect wide range of vectors as well as diverse mammalian host and their genetic diversity need to be investigated in detailed not only in Pakistan but also in various parts of the world to have more information about the prevalence and genotyping of this common apicomplexan parasite.

Analysis of the risk factor revealed that *Theileria ovis* infection varied between sheep breeds and the sampling sites. Parasitic infection was more common among small herds (Tables 2–4). Contrary to our observations, Arif et al. [28] found that sex, breed, herd size and herd composition were not associated with Theileria ovis infection in sheep enrolled from Fort Munro in Dera Ghazi Khan district in Punjab. Naz et al. [52] had also documented that sex, age and sampling seasons had no association with ovine and caprine theileriosis in Lahore district. On the other hand, Tanveer et al. [2] (2022) and Abid et al. [1] found that the presence of dogs with herds was a serious risk for the transmission of *T. ovis* in sheep enrolled from Rajanpur and Layyah districts respectively. This diverse epidemiological data warrants more large scale studies to be conducted in those areas of Pakistan that are unexplored for the presence of *Theileria ovis*.

During present investigation, we did not detect *Theileria lestoquardi* among the screened sheep. This absence or relatively low incidence of *Theileria lestoquardi* in sheep populations across Pakistan may be explained by the ecological characteristics and distribution of its main vector, *Hyalomma anatolicum*. Although this tick species is present in Pakistan, its population density, geographic range, and seasonal activity patterns may not align with optimal transmission dynamics in many regions of the country [1]. However, in Pakistan, other tick species like *Rhipicephalus* spp. are more abundant and widespread, particularly in arid and semi-arid zones. These species are more often associated with other *Theileria* types (e.g., *Theileria ovis* or *Theileria annulata*), reducing the chances of *Theileria lestoquardi* transmission [2].

## Conclusion

In conclusion, this study reports the prevalence of *Anaplasma* spp., *Anaplasma ovis*, and *Theileria ovis* in sheep from six districts of Punjab. Variations in infection rates were observed across different sampling districts and sheep breeds. The relatively high prevalence of these pathogens highlights the need for large-scale studies in Pakistan to better understand their dynamics at the human-animal interface. Implementation of effective preventive and control measures is essential to mitigate the economic impact of these infections on small ruminant production.

### ARRIVE guidelines

All the methods were performed in accordance with ARRIVE guidelines laws and regulations

## Statement of informed consent

Informed consent was obtained from livestock owners before including their animal in this study.

## Supporting information

**S1 Table. Primer sequences, targeted genes, annealing temperature and expected amplicon size for the detection of *Anaplasma* spp., *Anaplasma ovis, Theileria ovis* and *Theileria* lestoquardi in sheep blood samples that were collected during present study.**
(DOC)

## Author contributions

**Conceptualization:** Furhan Iqbal.

**Formal analysis:** Wajiha Shahzadi, Mughees Akbar, Arooj Ijaz, Arshad Hussain, Furhan Iqbal.

**Funding acquisition:** Ayman A. Swelum.

**Resources:** Wajiha Shahzadi, Arshad Hussain, Umair Shabbir.

**Software:** Ioannis A. Giantsis, Shakir Ullah, Maged A. AlGaradi, Adil Khan, Furhan Iqbal, Ayman A. Swelum.

**Supervision:** Furhan Iqbal.

**Writing – original draft:** Wajiha Shahzadi, Mughees Akbar.

**Writing – review & editing:** Arooj Ijaz, Arshad Hussain, Umair Shabbir, Ioannis A. Giantsis, Shakir Ullah, Maged A. AlGaradi, Adil Khan, Furhan Iqbal, Ayman A. Swelum.

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
