## [Decision Letter · Decision Letter 0]

Dear Dr. Iqbal,

Thank you for submitting your manuscript to PLOS ONE. After careful consideration, we feel that it has merit but does not fully meet PLOS ONE’s publication criteria as it currently stands. Therefore, we invite you to submit a revised version of the manuscript that addresses the points raised during the review process.

We look forward to receiving your revised manuscript.

Kind regards,

Abdelfattah Selim, PhD

Academic Editor

PLOS ONE

“The authors extend their appreciation to the Ongoing Research Funding Program (ORF-2025-971), King Saud University, Riyadh, Saudi Arabia, for funding this research.”

“The authors extend their appreciation to the Ongoing Research Funding Program (ORF-2025-971), King Saud University, Riyadh, Saudi Arabia, for funding this research”

“The authors extend their appreciation to the Ongoing Research Funding Program (ORF-2025-971), King Saud University, Riyadh, Saudi Arabia, for funding this research.”

Additional Editor Comments (if provided):

Reviewers' comments:

Reviewer's Responses to Questions

**Comments to the Author**

1. Is the manuscript technically sound, and do the data support the conclusions?

Reviewer #1: Yes

Reviewer #2: No

Reviewer #3: Yes

Reviewer #4: Yes

2. Has the statistical analysis been performed appropriately and rigorously?

Reviewer #1: Yes

Reviewer #2: No

Reviewer #3: Yes

Reviewer #4: Yes

3. Have the authors made all data underlying the findings in their manuscript fully available?

Reviewer #1: Yes

Reviewer #2: No

Reviewer #3: Yes

Reviewer #4: Yes

4. Is the manuscript presented in an intelligible fashion and written in standard English?

Reviewer #1: Yes

Reviewer #2: No

Reviewer #3: Yes

Reviewer #4: Yes

Reviewer #1: General Evaluation:

This manuscript presents a well-structured and scientifically sound study investigating the prevalence and genetic characterization of three major blood-borne pathogens Anaplasma spp., Anaplasma ovis, and Theileria ovis in sheep populations from six districts of Punjab, Pakistan. The topic is timely, relevant, and important for both veterinary health and livestock productivity, particularly in low-income rural regions where small ruminants play a key role in livelihoods and food security.

The authors have combined molecular techniques, including PCR amplification of 16S rRNA, msp4, and 18S rRNA genes, with phylogenetic analysis to assess pathogen diversity. This approach provides both prevalence data and evolutionary insights that are critical for understanding pathogen ecology and potential disease transmission dynamics in the region.

Strengths of the Manuscript:

Novelty and Regional Relevance:

The study fills an important knowledge gap regarding the epidemiology and molecular diversity of blood-borne pathogens in sheep from Pakistan, a country where such data are limited.

Comprehensive Sampling Strategy:

Sampling across six districts ensures geographic diversity and strengthens the generalizability of findings.

Robust Molecular and Phylogenetic Methods:

The use of highly conserved genetic markers for molecular identification and phylogenetic analysis is appropriate and yields informative results.

Relevant Risk Factor Analysis:

Associations between infection prevalence and factors such as herd size, breed, and geographical location are well-presented and contextualized with previous studies.

Well-Structured Discussion:

The discussion effectively compares local data with global findings, providing valuable insight into the pathogen's prevalence, diversity, and epidemiological trends.

Clear Conclusion and Implications:

The conclusion rightly emphasizes the need for continued surveillance, regional-scale phylogenetic studies, and the development of prevention strategies.

Minor Suggestions for Improvement:

Some grammatical and typographical corrections can improve readability. These are mostly minor and do not affect the scientific content. Figures and tables should be properly formatted according to journal guidelines. For clarity, brief information on the tick vectors present in the sampled regions could be added in future studies to strengthen the host-vector-pathogen relationship

Final Recommendation:

Accept with Minor Revisions

This manuscript makes a valuable contribution to veterinary parasitology and molecular epidemiology. The study is scientifically rigorous, the data are relevant, and the findings provide useful baseline information for future surveillance and control programs. I recommend acceptance after minor editorial revisions.

Reviewer #2: The manuscript reports molecular detection of four pathogens in sheep from six different regions of Pakistan. The authors collected the data and analyzed statistically for its significance regarding sex and geographic region of the sample collection. However, the manuscript suffers with several shortcomings:

1. The criteria for the selection of four pathogens and six geographic regions have not been stated. Why did authors ignore other pathogens and/or regions?

2. How did authors define ‘large herds’ versus ‘small herds?

3. What was the probability of the movement of herds from one region to other region?

4. How did authors calculated sample size? Why the number of samples differed in each district?

5. Method in 2.1 is not supported by the data. There wasn’t a structured epidemiological survey.

6. Method in 2.2 needs critical scrutiny. First, the authors did not use standard or type species as control. Also, the amplicon sizes are too small to assess the results. Have authors provided correct references for these primers? The reference with the doi 10.1016/j.parint.2017.09.002 led to a totally different paper. Other papers are not accessible.

7. How did authors confirm the information given in the lines 194-197.

8. On what basis authors decided to sequence some of the samples and not all? What was the criteria in deciding the number of samples to be sequenced?

9. Information in line 61 is contradictory to the information given in the abstract.

10. There are several language errors including spelling mistakes, formatting issues and redundancy.

Reviewer #3: The article titled "Molecular epidemiology and genetic diversity of Anaplasma and Theileria infections in sheep" by Shahzadi et al. and submitted by Dr. Furhan Iqbal is technically sound and contain important information related to anaplasmosis and theileriosis in sheep of Pakistan. Pakistan being agricultural country with over 70 percent population depends on livestock for their hands to mouth life, this study present important data about livestock parasitic diseases. I will recommend this article for acceptance based on the below mentioned minor edits.

1. Please remove extra information from abstract.

2. Please add more scholarly updated information about the tick borne diseases in Pakistan in the introduction section. This will help improve the rationale part of the introduction.

3. please also clearly mention the gap and need of this study in the last portion of introduction.

4. Please provide a clear map of the study using coordinates of the study area.

5. Results\are well interpreted and presented.

6. Please add some sentences in the discussion portion to make it more interested for the readers.

7. Please revise the conclusion portion based on the results of the study. Also tell the readers about what in future is possible regarding this topic.

Reviewer #4: Dear Authors,

I have reviewed the manuscript titled "Molecular epidemiology and genetic diversity of Anaplasma and Theileria infections in sheep" by Shahzadi et al. Below is my evaluation based on scientific content, language, and adherence to international standards.

Scientific Issues:

1. Title:

- The title is appropriate but could be more concise. Consider: "Molecular Epidemiology and Genetic Diversity of Anaplasma and Theileria spp. in Pakistani Sheep."

2. Abstract:

- The abstract lacks clarity in stating the novelty of the study. The significance of the findings should be emphasized.

- The phrase "phylogenetic evaluation of sp." is unclear and grammatically incorrect.

3. Keywords:

- "Molecular prevalence" is redundant; "Prevalence" suffices.

4. Introduction:

- The economic impact of these infections in Pakistan should be better justified with recent data.

- Some statements lack citations (e.g., "Pakistan is third in the World...").

5. Objectives:

- The aims are clear but could be more explicitly linked to gaps in existing literature.

6. Materials and Methods:

- The sampling strategy (e.g., random vs. convenience sampling) needs clarification.

- PCR conditions (annealing temperatures, cycle numbers) are missing.

- Statistical analysis: Clarify why Fischer’s exact test was used over chi-square for some comparisons.

7. Results:

- Table 1: The co-infection data is useful but should be presented more clearly.

8. Discussion:

- The discussion compares findings well but overstates some conclusions (e.g., genetic similarities with global strains without deeper analysis).

- The absence of Theileria lestoquardi should be discussed in the context of regional vector ecology.

Suggested Addition to the Discussion Section:

Add the two below citations to the reference list and the discussion.

https://doi.org/10.1016/j.smallrumres.2022.106617

https://doi.org/10.1007/s11250-021-03007-4

Two recent works by Noaman and colleagues provides valuable insights into the epidemiology of tick-borne pathogens in small ruminants.

Overall Assessment:

The manuscript presents valuable data on tick-borne pathogens in Pakistani sheep but requires minor revisions for clarity, grammatical accuracy, and methodological details. The results support the discussion and references need verification.

Recommendation: Accept after minor revisions.

**Do you want your identity to be public for this peer review?** For information about this choice, including consent withdrawal, please see our Privacy Policy

Reviewer #1: No

Reviewer #2: No

Reviewer #3: No

Reviewer #4: No

---

## [Author Response · Author response to Decision Letter 1]

23 Jun 2025

Editor’s comments

PONE-D-25-28992

Molecular epidemiology and genetic diversity of Anaplasma and Theileria infections in sheep

PLOS ONE

Dear Dr. Iqbal,

Thank you for submitting your manuscript to PLOS ONE. After careful consideration, we feel that it has merit but does not fully meet PLOS ONE’s publication criteria as it currently stands. Therefore, we invite you to submit a revised version of the manuscript that addresses the points raised during the review process.

e look forward to receiving your revised manuscript.

Kind regards,

Abdelfattah Selim, PhD

Academic Editor

PLOS ONE

Response to Editor’s comments

Dear Dr. Abdelfattah Selim, PhD

Academic Editor

PLOS ONE

I hope this letter will find you in the best of your conditions. We are highly grateful that you have kindly provided us an opportunity to revise our manuscript number PONE-D-25-28992 entitled “Molecular epidemiology and genetic diversity of Anaplasma and Theileria infections in sheep". We are also indeed thankful to you and the reviewer for the comments and suggestions that proved to be valuable feedback for us and helped us to significantly improve our manuscript. We believe that the reviewer’s comments can be entertained. So we have responded to all comments of the reviewer in a point wise manner in this letter and updated our manuscript as suggested by the reviewers. We have also extensively revised the manuscript in order to minimize the editing and typological errors and to make the manuscript more clear for the readers. The changes made in the revised manuscript can be seen through the TRACK CHANGES option of MS Word. We are submitting the updated and revised version to be considered for publication in your prestigious journal. Please feel free to contact us, if you have further questions.

Thanking you in anticipation.

With best regards

Furhan Iqbal

Corresponding author

Response

We have followed the instructions for authors.

“The authors extend their appreciation to the Ongoing Research Funding Program (ORF-2025-971), King Saud University, Riyadh, Saudi Arabia, for funding this research.”

Response

“The authors extend their appreciation to the Ongoing Research Funding Program (ORF-2025-971), King Saud University, Riyadh, Saudi Arabia, for funding this research”

“The authors extend their appreciation to the Ongoing Research Funding Program (ORF-2025-971), King Saud University, Riyadh, Saudi Arabia, for funding this research.”

Response

We are removing funding statement from manuscript.

Response

Ethical statement is only present in the declaration section.

Response

With reference to Fig 1, map was created by using the software QGIS system software. All the shape file having the administrative, provincial and district bounders was downloaded from the website DIVA-GIS, under the tab Free spatial Data. No need to seek consent from copyright holder as the map in Fig 1 was self-drawn. No Google Maps, Street View or Earth source has been used.

Response

Captions for your Supporting Information files are present at the end of our manuscript.

Response

Reference list has been revised carefully.

Reviewers' comments:

Reviewer's Responses to Questions

Reviewer #1:

General Evaluation:

This manuscript presents a well-structured and scientifically sound study investigating the prevalence and genetic characterization of three major blood-borne pathogens Anaplasma spp., Anaplasma ovis, and Theileria ovis in sheep populations from six districts of Punjab, Pakistan. The topic is timely, relevant, and important for both veterinary health and livestock productivity, particularly in low-income rural regions where small ruminants play a key role in livelihoods and food security. The authors have combined molecular techniques, including PCR amplification of 16S rRNA, msp4, and 18S rRNA genes, with phylogenetic analysis to assess pathogen diversity. This approach provides both prevalence data and evolutionary insights that are critical for understanding pathogen ecology and potential disease transmission dynamics in the region.

Strengths of the Manuscript:

Novelty and Regional Relevance:

The study fills an important knowledge gap regarding the epidemiology and molecular diversity of blood-borne pathogens in sheep from Pakistan, a country where such data are limited.

Comprehensive Sampling Strategy:

Sampling across six districts ensures geographic diversity and strengthens the generalizability of findings.

Robust Molecular and Phylogenetic Methods:

The use of highly conserved genetic markers for molecular identification and phylogenetic analysis is appropriate and yields informative results.

Relevant Risk Factor Analysis:

Associations between infection prevalence and factors such as herd size, breed, and geographical location are well-presented and contextualized with previous studies.

Well-Structured Discussion:

The discussion effectively compares local data with global findings, providing valuable insight into the pathogen's prevalence, diversity, and epidemiological trends.

Clear Conclusion and Implications:

The conclusion rightly emphasizes the need for continued surveillance, regional-scale phylogenetic studies, and the development of prevention strategies.

Response

Authors are indeed grateful for your kind words and in-depth review of our manuscripts. We have accepted all the suggested changes and made corrections accordingly. These changes have significantly improved our manuscript. Thank you.

Minor Suggestions for Improvement:

• Some grammatical and typographical corrections can improve readability. These are mostly minor and do not affect the scientific content.

Response

We have extensively revised the manuscript in order to minimize the editing and typological errors and to make the manuscript more clear for the readers. The changes made in the revised manuscript can be seen through the TRACK CHANGES option of MS Word.

• Figures and tables should be properly formatted according to journal guidelines.

Response

We have followed the instructions for the authors for table and figure formatting. Thanks for your kind reminder.

• For clarity, brief information on the tick vectors present in the sampled regions could be added in future studies to strengthen the host-vector-pathogen relationship

Response

Thanks for your suggestion. We have already provided some information regarding the ticks vectors that are associated with the screened pathogens. Following text is present in introduction section;

A variety of Anaplasma species infect sheep including Anaplasma marginale, Anaplasma ovis, Anaplasma phagocytophilum and Anaplasma capra [4]. A variety of ticks species belonging to Haemaphysalis , Ixodes and Rhipicephalus genera are transmit these bacteria to sheep and goats [9].

An apicomplexan parasites Theileria lestoquardi that is transmitted to small ruminants by the Hyalomma sp. ticks causes malignant ovine theileriosis that causes fever, cough, lethargy, lymphadenopathy and weight loss and can lead to mortality in case of severe infection [11].

• Final Recommendation:

Accept with Minor Revisions

This manuscript makes a valuable contribution to veterinary parasitology and molecular epidemiology. The study is scientifically rigorous, the data are relevant, and the findings provide useful baseline information for future surveillance and control programs. I recommend acceptance after minor editorial revisions.

Response

Authors are grateful for your kind recommendation.

Reviewer #2:

• The manuscript reports molecular detection of four pathogens in sheep from six different regions of Pakistan. The authors collected the data and analyzed statistically for its significance regarding sex and geographic region of the sample collection. However, the manuscript suffers with several shortcomings:

Response

Authors are indeed grateful for your kind words and in-depth review of our manuscripts. We have accepted all the suggested changes and made corrections accordingly. These changes have significantly improved our manuscript. Thank you.

1. The criteria for the selection of four pathogens and six geographic regions have not been stated. Why did authors ignore other pathogens and/or regions?

Response

Thanks for your query. We have mentioned in the introduction as well as in the discussion section that limited studies in specific areas have been conducted in Pakistan to report the prevalence of Anaplasma and Theileria species in Pakistani sheep and most of the districts are unexplored. To address this gap, sheep were enrolled from six districts in Punjab (Pakistan) and their blood samples were screened for the DNA of Anaplasma sp., Anaplasma ovis, Theileria ovis and Theileria lestoquardi through PCR and DNA sequencing approach. The pathogens selected for the screening are frequently reported in local small ruminants but sheep from five of the six districts has never been screened for these bacteri

---

## [Decision Letter · Decision Letter 1]

Molecular epidemiology and genetic diversity of Anaplasma and Theileria infections in sheep

PONE-D-25-28992R1

Dear Dr. Iqbal,

We’re pleased to inform you that your manuscript has been judged scientifically suitable for publication and will be formally accepted for publication once it meets all outstanding technical requirements.

Kind regards,

Abdelfattah Selim, PhD

Academic Editor

PLOS ONE

Additional Editor Comments (optional):

The authors have thoroughly addressed all reviewer comments and significantly improved the manuscript. I recommend acceptance in its current form.

Reviewers' comments:

Reviewer's Responses to Questions

**Comments to the Author**

Reviewer #1: All comments have been addressed

Reviewer #3: All comments have been addressed

Reviewer #4: All comments have been addressed

2. Is the manuscript technically sound, and do the data support the conclusions?

Reviewer #1: Yes

Reviewer #3: Yes

Reviewer #4: Yes

3. Has the statistical analysis been performed appropriately and rigorously?

Reviewer #1: Yes

Reviewer #3: Yes

Reviewer #4: Yes

4. Have the authors made all data underlying the findings in their manuscript fully available?

Reviewer #1: Yes

Reviewer #3: Yes

Reviewer #4: Yes

5. Is the manuscript presented in an intelligible fashion and written in standard English?

Reviewer #1: Yes

Reviewer #3: Yes

Reviewer #4: Yes

Reviewer #1: Thank you for your thorough and thoughtful revision of the manuscript. I have carefully reviewed the revised version and the authors’ responses to the previous comments. I am pleased to note that all the minor concerns raised in the initial review have been fully addressed.

The revised manuscript demonstrates improved clarity, coherence, and scientific rigor. The additional data and explanations provided have significantly strengthened the overall quality and impact of the work. The methods are now clearly described, the figures and tables are appropriate, and the conclusions are well-supported by the results.

Based on the comprehensive revisions and the satisfactory responses to all reviewer comments, I recommend the manuscript be accepted for publication in its current form.

Reviewer #3: All my comments on the previous version is well addressed. Hence, i recommend acceptance of this article.

Reviewer #4: Corrections are acceptable. The authors made all the proofreading and suggestions. Now, the manuscript can be considered for publication in the PLOS ONE as a Research Article.

**Do you want your identity to be public for this peer review?** For information about this choice, including consent withdrawal, please see our Privacy Policy

Reviewer #1: No

Reviewer #3: No

Reviewer #4: No

---

## [Editor Report · Acceptance letter]

PONE-D-25-28992R1

PLOS ONE

Dear Dr. Iqbal,

I'm pleased to inform you that your manuscript has been deemed suitable for publication in PLOS ONE. Congratulations! Your manuscript is now being handed over to our production team.

Kind regards,

on behalf of

Prof Abdelfattah Selim

Academic Editor

PLOS ONE